# Two-dimensional infrared-Raman spectroscopy as a probe of water's tetrahedrality

Tomislav Begušić [1] & Geoffrey A. Blake [1,2]

Two-dimensional spectroscopic techniques combining terahertz (THz), infrared (IR), and visible pulses offer a wealth of information about coupling among vibrational modes in molecular liquids, thus providing a promising probe of their local structure. However, the capabilities of these spectroscopies are still largely unexplored due to experimental limitations and inherently weak nonlinear signals. Here, through a combination of equilibrium-nonequilibrium molecular dynamics (MD) and a tailored spectrum decomposition scheme, we identify a relationship between the tetrahedral order of liquid water and its two-dimensional IR-IR-Raman (IIR) spectrum. The structure-spectrum relationship can explain the temperature dependence of the spectral features corresponding to the anharmonic coupling between low-frequency intermolecular and high-frequency intramolecular vibrational modes of water. In light of these results, we propose new experiments and discuss the implications for the study of tetrahedrality of liquid water.

Understanding the dynamical structure of liquid water is paramount to a number of chemical and biological processes. In particular, the tetrahedral ordering of molecules, stemming from the directionality of hydrogen bonds, has been proposed as the origin of water's anomalous behavior in the liquid phase[1]. However, the tetrahedral structure of water has also been contested both computationally[2] and experimentally[3]. To date, most of our microscopic, structural information about liquid water comes from molecular dynamics (MD) simulations, which depend strongly on the choice of electronic structure theory or force field parametrization. In contrast, experimental tools capable of studying the local arrangement of water molecules remain scarce[4]. For example, common techniques that probe the structure of liquids, such as X-ray and neutron scattering, typically report on highly time-averaged quantities and can be accurately reproduced with very different structural motifs[2,5–7]. Vibrational spectroscopy, such as IR absorption and Raman scattering, offers complementary information about the strength of intermolecular hydrogen bonds, which depends on instantaneous, local arrangement of water molecules[8]. Different spectral regions have been probed to investigate intermolecular hydrogen-bond bending and stretching modes (up to 300 cm$^{-1}$), frustrated rotational (librational) modes (400–1000 cm$^{-1}$), intramolecular bending (1650 cm$^{-1}$) and stretching

(3000–3800 cm$^{-1}$) modes, as well as combination bands between intermolecular and intramolecular modes. Reported experiments and simulations range from high-resolution spectroscopy on molecular clusters[9–12] to studies of interfacial[13–15] or bulk water[16–25]. Recently, a joint experimental and computational study[26,27] of the temperature dependence of the Raman spectrum of liquid water revealed a clear structure-spectrum relationship between the local tetrahedral order parameter[28], a measure of structuring of liquid water, and frequency shifts and intensities of spectral peaks. In fact, the temperature dependence of the spectrum was, in this way, fully explained as a change in the thermal distribution of the tetrahedral order parameter. Yet, conventional, steady-state spectroscopy of liquids typically produces broad, unresolved spectral features and misses dynamical information.

Time-resolved and two-dimensional vibrational spectroscopies have emerged in the past years as probes sensitive to local water coordination. Even here, however, the long-standing tool of two-dimensional infrared spectroscopy (2DIR)[4,29,30], which measures the interaction of light with intramolecular, high-frequency modes, provides only an indirect probe of intermolecular dynamics. 2DIR has been successfully applied to analyze coupling among high-frequency modes in ice[31] and in proteins[32], but also to understand the vibrational

[1]Division of Chemistry and Chemical Engineering, California Institute of Technology, Pasadena, CA 91125, USA. [2]Division of Geological and Planetary Sciences, California Institute of Technology, Pasadena, CA 91125, USA. ✉e-mail: tbegusic@caltech.edu; gab@caltech.edu

relaxation mechanisms in liquids[33]. For example, several 2DIR studies on liquid water and ice speculated that mechanical anharmonic coupling to intermolecular modes contributes to the relaxation of the OH stretch[34–36]. In addition, non-Condon effects, which are related to electrical anharmonic coupling between high- and low-frequency modes, were shown to affect the 2DIR spectra of OH stretching mode in liquid water[37]. To target the low-frequency modes directly, a number of hybrid spectroscopic techniques have been proposed, involving different sequences of THz, IR, and visible pulses, such as the THz-THz-Raman[38–41], THz-Raman-THz, Raman-THz-THz[42–45], and THz-IR-Raman (also called THz-IR-visible[46,47] or TIRV). Their development was enabled by the recent advances in the generation of strong THz pulses that are needed to induce a nonlinear light-matter interaction[48]. The THz-THz-Raman and TIRV methods are related to other two-dimensional IR-Raman techniques, namely the two-dimensional IR doubly vibrationally enhanced and IR-IR-visible sum-frequency generation spectroscopies[49–54]. For example, TIRV experiments have revealed unambiguous spectral signatures of coupling between the intramolecular O-H stretch and intermolecular hydrogen-bond bending and stretching modes[46]. Theoretical simulations by Ito and Tanimura[55] predicted such spectral features and assigned them to both mechanical and electrical anharmonic coupling between the said vibrational modes. Similarly, THz-THz-Raman spectroscopy recently revealed signatures of anharmonic coupling between phonons of ionic solid LiNbO$_3$[56]. Finally, Raman-THz-THz and THz-Raman-THz spectroscopies have been used to study the inhomogeneity of liquid water and aqueous solutions[57,58], as well as the coupling among intermolecular and intramolecular modes in liquid and solid bromoform[59,60]. Even so, due to limited availability of efficient THz emitter materials, not all frequencies have been covered by the reported techniques. Specifically, most two-dimensional hybrid THz-Raman spectroscopies of liquid water targeted hydrogen-bond bending and stretching modes, i.e., frequencies up to 400 cm$^{-1}$, leaving the water librational dynamics largely unexplored.

Here, we aim to provide new insights into the capabilities of two-dimensional hybrid IR-Raman vibrational spectroscopies. To this end, we study the temperature dependence of the two-dimensional IR-IR-Raman (IIR) spectrum, which is given by the double Fourier (or sine) transform of an appropriate third-order, two-time response function. Since the computational model involves all vibrational modes of the

system, the response function covers a broad range of frequencies, which, in practice, can be mapped out only through separate experiments. To date, only the TIRV frequency region has been experimentally measured, although its temperature dependence was not studied. Second, following ref. 26, we then establish a structure-spectrum relationship by separating the spectral contributions from molecules exhibiting low or high tetrahedral coordination. Third, to justify the molecular dynamics (MD) results at low temperatures, we analyze whether nuclear quantum effects are discernible in the two-dimensional IIR spectrum.

## Results
### Theoretical model
We simulated the IIR response function[40,55]

$$R(t_1,t_2) = -\frac{1}{\hbar^2}\text{Tr}\{[[\hat{\Pi}(t_1+t_2),\hat{\mu}(t_1)],\hat{\mu}(0)]\hat{\rho}\} \quad (1)$$

using the equilibrium-nonequilibrium MD approach, in which the quantum-mechanical trace is replaced by a classical average[61–64]

$$R^{\text{MD}}(t_1,t_2) = \frac{\beta}{\varepsilon}\langle[\Pi(q_{+,t_2}) - \Pi(q_{-,t_2})]\dot{\mu}(q_{-t_1})\rangle, \quad (2)$$

where $\hat{\rho}$ is the thermal density operator, $\hat{\mu} = \mu(\hat{q})$ is the dipole moment operator, and $\hat{\Pi} = \Pi(\hat{q})$ is the polarizability. $\mathbf{q}_t$ denotes the position vector of a classical trajectory initiated at $(\mathbf{q}_0, \mathbf{q}_0)$, whereas $\mathbf{q}_{\pm,t}$ corresponds to the initial conditions $(\mathbf{q}_{\pm,0} = \mathbf{q}_0, \mathbf{p}_{\pm,0} = \mathbf{p}_0 \pm \varepsilon\mu'(\mathbf{q}_0)/2)$ after an instantaneous interaction with the electric field. $\varepsilon$ is a free parameter in the calculations and corresponds to the magnitude of the external electric field integrated over the short interaction time. For sufficiently small $\varepsilon$, the thermal average of Eq. (2) is linear in $\varepsilon$, meaning that the time-dependent response function $R^{\text{MD}}(t_1, t_2)$ is, as expected, independent of its exact value[63,65]. The two-dimensional spectra are computed through a double sine transform[42,55]

$$R(\omega_1,\omega_2) = \int_0^\infty \int_0^\infty R(t_1,t_2)\sin(\omega_1 t_1)\sin(\omega_2 t_2)dt_1 dt_2. \quad (3)$$

To model water, we used a flexible, point-charge qTIP4P/F force field[66], which has been well studied for spectroscopic simulations[25] and benchmarked against a number of experimental thermodynamic properties of water, including the radial distribution functions, dielectric constant, density, and melting point. As a point-charge force field, the model is computationally efficient, which is needed for statistically averaging two-time response functions. In addition, because it is flexible, we could study all vibrational degrees of freedom, including high-frequency intramolecular modes. To allow for nonlinear dependence of the dipole moments and polarizabilities on nuclear coordinates, we employed the truncated dipole-induced-dipole (DID) model. Following Hamm[44], each water molecule was amended with permanent anisotropic polarizability, which was used for the evaluation of the induced contributions to the dipoles and polarizabilities. In contrast to the rigid water simulations of ref. 44, here the permanent polarizability was an explicit function of intramolecular degrees of freedom, according to ref. 67. The introduced induced dipole and polarizability effects were not used in the evaluation of the forces, which were computed according to the original, non-polarizable qTIP4P/F model. We note that more advanced models for the potential energy, dipoles, and polarizabilities of liquid water exist and have been used in the simulation of one- and two-dimensional spectroscopies[19,22,24,26,44,68]. However, while some of them were not readily available, others considered only rigid water molecules or were fitted to experiments using classical MD simulations, which would prevent us from accurately exploring nuclear quantum

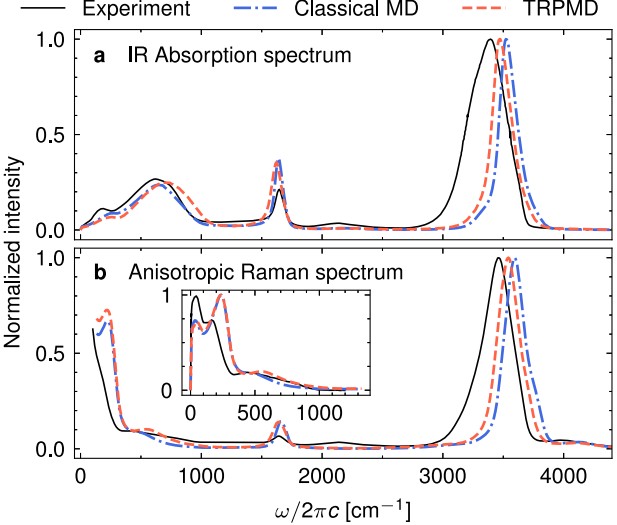

**Fig. 1 | Steady-state spectra of water.** IR absorption (**a**) and anisotropic Raman (**b**) spectra of liquid water at 300 K simulated with the classical MD and TRPMD, compared with the experiments of refs. 16,27,91. The inset in panel **b** compares the low-frequency part of the simulated and experimental anisotropic Raman spectra.

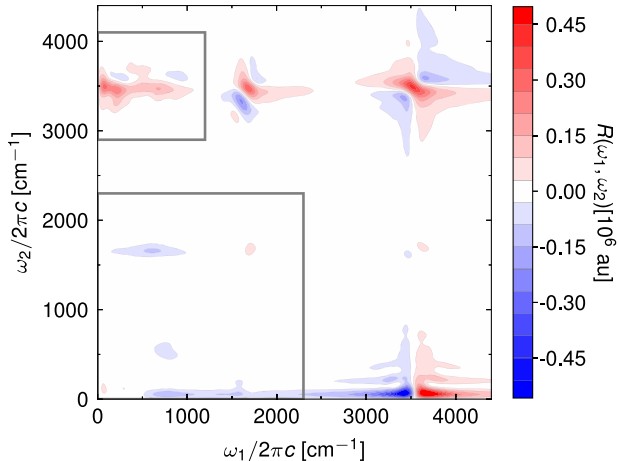

**Fig. 2 | Two-dimensional IIR spectrum of liquid water simulated with classical MD at 300 K.** Grey squares indicate the regions of the spectrum that we studied in more detail, namely the region of TIRV spectroscopy and the low-frequency region that covers intermolecular modes and the intramolecular bending mode.

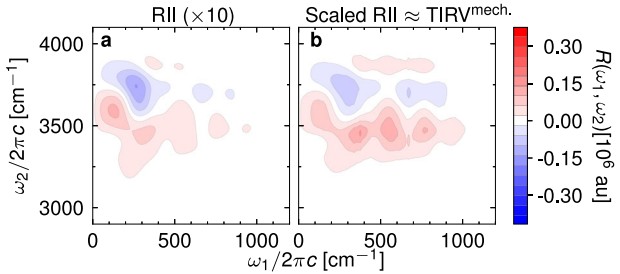

**Fig. 3 | Analysis of the mechanical anharmonic coupling. a** RII spectrum with $\mu^{ind} = 0$ simulated using classical MD at 300K (multiplied by 10 so that the spectral features can be seen on the same scale as in panel **b**). **b** The same spectrum multiplied by the frequency-dependent scaling factor derived in Supplementary Eq. (11), which approximates the mechanical anharmonic coupling contribution to the TIRV spectrum.

effects. In Fig. 1, we show that the employed force field combined with our induced dipole and polarizability models can reproduce the main features of the experimental IR absorption and anisotropic Raman spectra of liquid water. Additional details about the model and MD simulations can be found in the Methods section.

Most of the results presented below rely on the validity of the classical MD approach, which neglects the quantum-mechanical properties of atomic nuclei. However, due to the presence of light hydrogen atoms, their effect might not be negligible[66]. Although such nuclear quantum effects on one-dimensional IR and Raman spectra have been well studied using approximate but reliable classical-like methods, no tools similar to the equilibrium-nonequilibrium MD have been available to study nuclear quantum effects on two-dimensional IR-Raman spectra[69-71]. Recently, we have developed a new ring-polymer MD (RPMD) approach[65], which can simulate, at least approximately, the nuclear quantum effects on the two-dimensional IIR spectra. Briefly, the RPMD method[72] replaces the original quantum-mechanical problem by an extended classical system consisting of $N$ replicas (beads) of the original system connected by harmonic springs. For a given potential energy surface, the extended classical system in the limit of $N \to \infty$ reproduces the exact quantum-mechanical thermal distribution of nuclear degrees of freedom, while if $N = 1$, RPMD reduces to classical MD. In our simulations, we used $N = 32$, which is sufficiently large for liquid water in the studied temperature range[25]. Since RPMD is known to suffer from the spurious resonance issue,

where unphysical peaks due to artificial harmonic springs appear in the spectra, we employed its thermostatted version (TRPMD)[73,74]. IR absorption and anisotropic Raman spectra simulated with TRPMD are presented in Fig. 1.

## Two-dimensional IIR spectrum of liquid water

Figure 2 shows the full two-dimensional IIR spectrum of liquid water simulated at 300 K. Apart from the spectral features along the $\omega_1 = \omega_2$ diagonal, which appear due to mechanical or electrical anharmonicity of individual vibrational modes, the spectrum contains off-diagonal peaks that correspond to coupling among different vibrations. Overall, the spectrum qualitatively agrees with the simulation of ref. 55, with the main difference in the relative intensities of different spectral regions, which can depend strongly on the details of the potential energy, dipole, and polarizability surfaces. In the following, we focus on the two frequency regions indicated by the grey rectangles. One region targets the coupling between intermolecular modes ($0\ cm^{-1} < \omega_1 < 1200\ cm^{-1}$) and the intramolecular O-H stretch mode ($2900\ cm^{-1} < \omega_2 < 4100\ cm^{-1}$), whereas the other covers all low-frequency, intermolecular modes, and the intramolecular bend mode. Model simulations of ref. 55. (Supplementary Discussion 1 and Supplementary Fig. 3) imply that the complex spectral lineshape of the former, which we will refer to as the TIRV region[46], are a product of interplay between mechanical and electrical anharmonicity. Namely, mechanical anharmonicity leads to a shape comprising a positive and a negative lobe above and below the central $\omega_2$ frequency, whereas electrical anharmonicity produces an approximately symmetric feature. To verify this interpretation, we would ideally construct approximate dipole and polarizability models that depend linearly on atomic coordinates, which would allow us to study the mechanical anharmonic coupling pathways directly. This can be easily done for the dipole moments by neglecting the induced part because the permanent molecular dipole is a linear function of coordinates (see Methods). Unfortunately, the same cannot be achieved for polarizability because both permanent and induced parts are nonlinear in atomic coordinates. For this reason, we consider an alternative, Raman-IR-IR (RII) pulse sequence, in which the polarizability is responsible for the first interaction with the external electric field. Assuming weak electrical and mechanical anharmonic coupling, it can be shown that the nonlinearity in the first interaction does not contribute to the RII spectrum (Supplementary Discussion 2). Therefore, the RII spectrum with permanent (i.e., linear) dipole moments (shown in Fig. 3a) consists only of mechanical anharmonic coupling pathways. Because the Raman response in the THz frequency range is weak, the corresponding RII spectrum exhibits roughly an order of magnitude lower intensity than the TIRV spectrum. For comparison with the full TIRV spectrum, the simulated RII spectrum must be appropriately scaled (Fig. 3b) along the frequency axes (Supplementary Eq. (9)). The result agrees with the proposed interpretation that the nodal shape of the spectrum results from mechanical anharmonic coupling pathways.

In the low-frequency region of interest, we observe a strong peak at about ($600\ cm^{-1}$, $1650\ cm^{-1}$) due to anharmonic coupling between librations ($400$–$1000\ cm^{-1}$) and the intramolecular bending mode[75]. The same coupling mechanism is responsible for the combination transition at around $2150\ cm^{-1}$, also known as the "association" band[9,20], in the one-dimensional spectra. Interestingly, this combination band is not captured in our model (Fig. 1), even though the corresponding two-dimensional spectral feature clearly appears in the IIR spectrum. This discrepancy can be explained by the fact that the fundamentals of the one-dimensional spectra follow harmonic selection rules, while the peaks in the two-dimensional spectra appear solely due to the anharmonic excitation pathways. Therefore, the combination bands in the one-dimensional spectra can be orders of magnitude weaker than the fundamentals and still exhibit strong off-diagonal peaks in the two-dimensional spectra. Indeed, we see that the diagonal

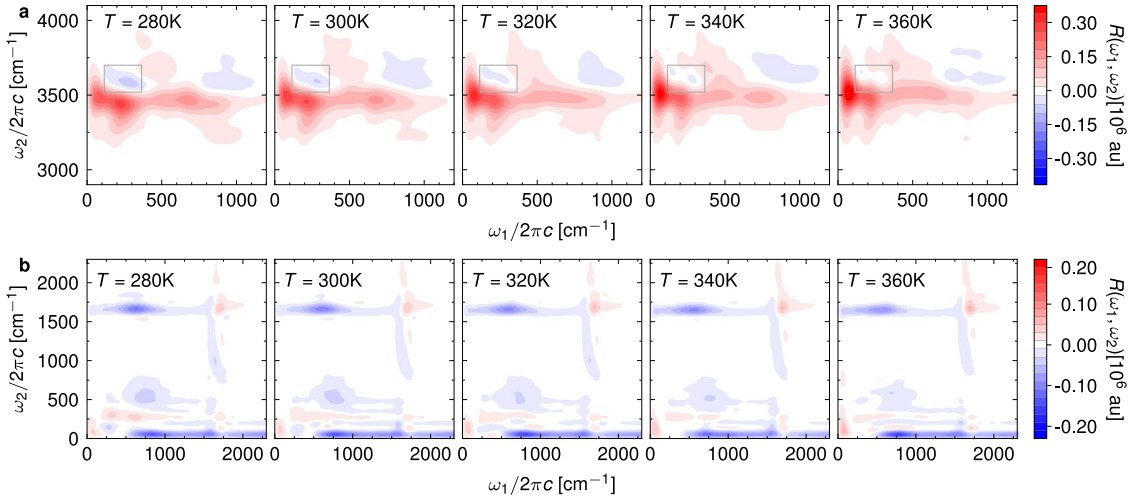

**Fig. 4 | Temperature dependence of the IIR spectrum.** Two-dimensional IIR spectra of liquid water simulated with classical MD at different temperatures. **a** TIRV part of the spectrum. The grey squares highlight the peak around (250 cm⁻¹, 3600 cm⁻¹). **b** Low-frequency part of the spectrum.

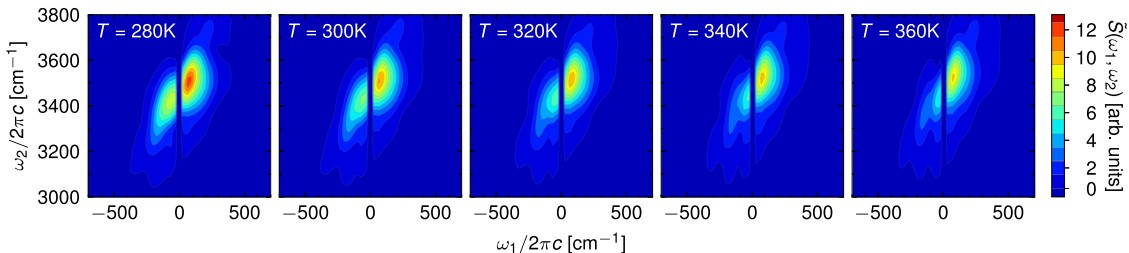

**Fig. 5 | Temperature dependence of the TIRV spectrum related to the experiment of ref. 47.** Two-dimensional TIRV spectra of liquid water at different temperatures, as calculated using Eq. (4) and time-dependent response functions $R(t_1, t_2)$ equal to those used in Fig. 4.

peak at around (1650 cm⁻¹, 1650 cm⁻¹) is much lower in intensity than the off-diagonal libration-bending peak, implying that the anharmonicity within the bending mode is weaker than its anharmonic coupling to the intermolecular librations. Let us note that other force fields and dipole/polarizability models also heavily underestimate or fail to reproduce the combination band at 2150 cm⁻¹ [19,21,55,76], unlike the ab initio approaches, which appear to systematically reproduce it [26,77,78]. Since the force field we used reproduces the structural and dynamical properties of liquid water rather accurately [66], we tentatively assign the absence of this combination band in the simulated spectra (Fig. 1) to the limitations of our dipole and polarizability models.

### Temperature dependence of the IIR spectrum

We now turn to the temperature dependence of the two IIR spectral regions (Fig. 4). The TIRV part of the spectrum experiences several changes as the temperature is increased from 280 K to 360 K. At higher temperatures, the peaks are blue-shifted along $\omega_2$, which agrees with the observed trends in the experimental Raman spectra and simulated vibrational density of states [26]. In addition, the shape of the spectral peaks changes drastically. The most prominent change in the spectrum is the disappearance of the negative peak around (250 cm⁻¹, 3600 cm⁻¹). This indicates that the electrical anharmonicity contribution to the spectrum, marked by a strongly symmetric lineshape, increases compared to that of the mechanical anharmonicity. Unlike the frequency shift, this feature cannot be accessed through steady-state spectroscopy. Finally, we note that a peak around (400 cm⁻¹, 3700 cm⁻¹) appears at elevated temperature, while the other peak at about (700 cm⁻¹, 3400 cm⁻¹) almost disappears. Interestingly, both of

these coupling terms have been discussed as possible origins of the libration-stretch combination band appearing in the anisotropic Raman spectrum of liquid water at 4100 cm⁻¹ [26,79]. The same combination band has also been shown to play an important role in second-order vibrational sum-frequency spectroscopy of interfacial water [15].

The low-frequency region also exhibits strong temperature dependence. For example, the results (see also difference spectra in Supplementary Fig. 5) clearly indicate a red shift of the libration-bend peak along $\omega_1$ with increasing temperature. This is a straightforward consequence of the equivalent frequency shift found in the linear IR absorption spectrum (Supplementary Fig. 6), which also agrees with the experimentally observed trend [17]. Furthermore, the two-dimensional IIR spectrum at 280 K contains peaks close to diagonal, around $\omega_1 = \omega_2 = 250$ cm⁻¹, due to anharmonicity of the hydrogen-bond stretching modes. These seem to progressively disappear at elevated temperatures. Importantly, this change in intensity could not be simply predicted from the one-dimensional spectra, which show little change in the intensity of the 250 cm⁻¹ band. We note, however, that the features in this congested spectral region could be affected by the short times available from our simulations and by artificial broadening. Longer simulations are possible with rigid water models, which have been used to study explicitly the intermolecular modes and corresponding two-dimensional THz-Raman signals in the time domain [44,62].

The TIRV spectral region has been studied experimentally [46,47]. However, the experiments could only measure an absolute value Fourier transform spectrum, which is quite different from the real sine transform discussed here and in ref. 55. Specifically, the experimental spectrum reported in Fig. 8 of ref. 47 is related to the time-dependent

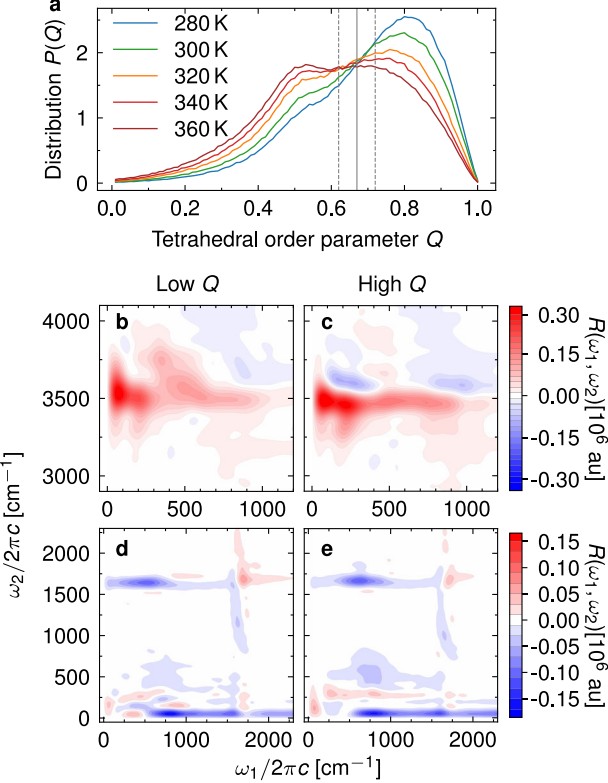

**Fig. 6 | Effect of tetrahedral order on the IIR spectra. a** Distribution of the local tetrahedral order parameter of liquid water as a function of temperature. The isosbestic value is indicated by the solid, grey line. Two dashed, grey lines specify the region excluded from subsequent two-dimensional IIR spectra simulations ($0.62 < Q < 0.72$). Two-dimensional IIR spectra of water molecules with high ($Q > 0.72$, **c**, **e**) or low ($Q < 0.62$, **b**, **d**) tetrahedral order parameter, simulated with the classical equilibrium-nonequilibrium MD approach at 320 K.

response function by

$$\tilde{S}(\omega_1, \omega_2) = |[S(\omega_1, \omega_2) + S(\omega_2 - \omega_1, \omega_2)]E_{THz}(\omega_1)E_{IR}(\omega_2 - \omega_1)|, \quad (4)$$

$$S(\omega_1, \omega_2) = \int_0^\infty \int_0^\infty R(t_1, t_2)e^{-i\omega_1 t_1}e^{-i\omega_2 t_2}dt_1 dt_2, \quad (5)$$

where $E_{THz}(\omega)$ and $E_{IR}(\omega)$ are the THz and IR electric fields obtained as square roots of the pulse intensity spectra presented in Fig. 2b, c of ref. 46. The main differences between our simulation and the experiment of ref. 47 arise due to the limitations of our model, namely the narrow and blue-shifted OH stretch (Fig. 1). Furthermore, the experimental spectrum exhibits additional spectral features that cannot be explained with Eq. (4) and the pulse shapes we used because they fall outside of the bandwidth of the instrument response function. Unlike the sine-transform spectra shown in Fig. 4, Fourier-transform spectra convolved with external electric fields (Fig. 5) only become less intense with increasing temperature but show no interesting spectral change. Therefore, an accurate determination of the full response function will be needed to experimentally measure the spectral features appearing in the sine-transform TIRV spectra. Fortunately, a scheme that could achieve this goal has been recently applied to the TIRV spectrum of liquid dimethyl sulfoxide[80], demonstrating that similar experiments on water are within reach.

## Spectral signatures of water's tetrahedrality in the IIR spectrum
To understand the temperature-dependent spectral features, we follow the work of Morawietz et al.[26], where the temperature effects were analyzed in relation to the local structuring around individual water

molecules. This can be quantified by the local tetrahedral order parameter $Q$[81–83], which measures how the arrangement of the four neighboring water molecules deviates from the ideal tetrahedral arrangement around the central one. By convention, $Q = 1$ for a perfect tetrahedral arrangement, while $Q = 0$ for an ideal gas. The distribution of $Q$ at thermal equilibrium exhibits a strong temperature dependence and a clear isosbestic point at $Q \approx 0.67$ (Fig. 6a). In the studied temperature range, the bimodal distribution can be decomposed into two temperature-independent components whose populations change with temperature[26]. However, as shown by Geissler[84], the appearance of an isosbestic point does not necessarily imply that water is a heterogeneous mixture. More recently, the increased population of low-$Q$ water molecules at higher temperatures has been assigned to the appearance of neighboring molecules in the interstitial position between the first and second solvation shells[83].

To directly correlate the local tetrahedral order parameter with the spectral features observed in the two-dimensional IIR spectra, we decomposed the spectrum into contributions of individual molecules (Supplementary Discussion 3). Then, we simulated the spectra (Fig. 6b–e) originating predominantly from the molecules with either high ($Q > 0.72$) or low ($Q < 0.62$) local tetrahedral order parameters. The two IIR spectra align remarkably well with the observed temperature-dependent changes. Specifically, the high-order TIRV spectrum (Fig. 6c) exhibits a clear mechanical anharmonic coupling feature with a positive and a negative lobe at $\omega_1 \approx 250\ cm^{-1}$, whereas the corresponding low-order spectrum contains a strong symmetric peak, indicating electrical anharmonic coupling between the hydrogen-bond modes and O-H stretch. In addition, the libration-stretching peak at $(400\ cm^{-1}, 3700\ cm^{-1})$ appears almost exclusively in molecules with low tetrahedral order. Similarly, the libration-bending peak in the low-frequency part the spectrum (Fig. 6d, e) appears at lower $\omega_1$ frequencies for low $Q$. Overall, the temperature dependence of the IIR spectra can be almost exclusively assigned to the changes in the distribution of the local tetrahedral order parameter and the effect of the local structure on the spectral features. Therefore, TIRV spectroscopy expanded into the water libration frequency range, i.e., with $\omega_1$ covering up to $\approx 800\ cm^{-1}$, could provide an alternative probe of the local molecular structure in liquid water and aqueous solutions. Importantly, the spectral changes in IIR are more drastic, and therefore more sensitive to local ordering, than the frequency shifts observed in conventional, one-dimensional IR and Raman spectroscopies.

This result is consistent with other experimental observations and theoretical models. Namely, it is known that as the tetrahedral order parameter increases, the hydrogen-bond stretching mode frequency increases and the OH stretching frequency decreases[26], which can be interpreted in terms of a growing mechanical anharmonic coupling strength between the two modes. A more detailed analysis is possible based on the work of Auer and Skinner[85], who studied the dependence of the intramolecular stretching frequency and the corresponding dipole moment derivative on the electric field $E$ generated by the surrounding water molecules at the hydrogen atom of the central molecule and projected along the OH bond. They found that the frequency can be fit to a quadratic function of this electric field, while the dipole moment derivative is approximately linear in $E$:

$$\omega_{OH} = \omega_{OH}^{(0)} - \omega_{OH}^{(1)}E - \omega_{OH}^{(2)}E^2, \quad (6)$$

$$\mu'_{OH} = \mu'^{(0)}_{OH} + \mu'^{(1)}_{OH}E, \quad (7)$$

where all parameters $\omega_{OH}^{(\alpha)}$ and $\mu'^{(\alpha)}_{OH}$ are positive and defined in Table I. of ref. 85. More recent models[86,87] included a very weak quadratic term for the dipole moment as well, which can be neglected within the typical range of values of the electric field. For us, the relevant anharmonic coupling quantities are the derivatives of $\omega_{OH}$ and $\mu'_{OH}$

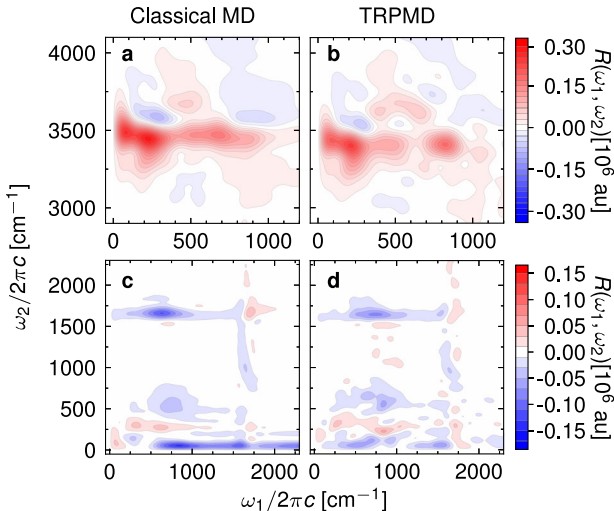

**Fig. 7 | Nuclear quantum effects on the IIR spectrum.** Two-dimensional IIR spectra of liquid water at 280 K, simulated with equilibrium-nonequilibrium MD (**a**, **c**) and TRPMD (**b**, **d**).

with respect to an intermolecular hydrogen-bond mode $q_{HB}$, $\partial\omega_{OH}/\partial q_{HB} = -(\omega_{OH}^{(1)} + 2\omega_{OH}^{(2)}E)\partial E/\partial q_{HB}$ and $\partial\mu'_{OH}/\partial q_{HB} = \mu_{OH}^{\prime(1)}\partial E/\partial q_{HB}$. From these equations, we see that the mechanical anharmonic coupling, determined by $|\partial\omega_{OH}/\partial q_{HB}|$, grows with the electric field, whereas the electrical anharmonic coupling, proportional to $|\partial\mu'_{OH}/\partial q_{HB}|$, has no explicit dependence on $E$. In ref. 83, it has been shown that $E$ correlates positively with the tetrahedral order parameter $Q$, which further validates the interpretation of our two-dimensional spectroscopy simulations, i.e., stronger mechanical anharmonic coupling in high-$Q$ molecules.

### Nuclear quantum effects on the two-dimensional IIR spectrum

Finally, we report the first TRPMD simulation of the two-dimensional IIR spectrum of liquid water and compare it with the MD simulation in Fig. 7. Some nuclear quantum effects can be easily explained in terms of the differences between one-dimensional spectra (Supplementary Fig. 7). Namely, the red-shifted O-H stretch band in the TRPMD steady-state spectra reflects in a red shift along $\omega_2$ of the spectral features in the TIRV spectrum (Fig. 7a, b). Furthermore, the libration-stretch peak shifts from $\omega_1 \approx 700\,\text{cm}^{-1}$ in the MD spectrum to $\omega_1 \approx 800\,\text{cm}^{-1}$ in TRPMD, which aligns with the differences between MD and TRPMD IR absorption spectra. In the low-frequency part of the spectrum (Fig. 7c, d), the libration-bend peak is analogously shifted along $\omega_1$. However, some features cannot be understood from comparison with the one-dimensional spectra, such as the absence of the strong negative feature at ($\omega_1 \approx 750\,\text{cm}^{-1}$, $\omega_2 \approx 50\,\text{cm}^{-1}$) in TRPMD or the change in the spectral lineshapes in the TIRV region at $\omega_1 > 400\,\text{cm}^{-1}$. Nevertheless, these differences are not sufficiently large to alter the above conclusions based on classical MD simulations.

## Discussion

To conclude, we have presented MD and TRPMD simulations of the two-dimensional IIR spectra of liquid water. By analyzing the temperature dependence of the spectrum, we have identified features that report on the degree of local tetrahedral ordering in the first coordination shell. Further computational work is needed to confirm whether IIR spectra of related systems, such as aqueous solutions, alcohols, or ice, can be broadly mapped to the local tetrahedral order parameter. Our work demonstrates that the temperature dependence of such two-dimensional spectra can be used in combination with computational modeling to gain insight into the structure of complex liquids.

Furthermore, the simulations presented here imply that the contributions of mechanical and electrical anharmonic coupling change as a function of local tetrahedrality, which cannot be studied with conventional, one-dimensional spectroscopy. Specifically, the electrical anharmonicity dominates at higher temperatures, where the tetrahedral order parameter is low, whereas the mechanical coupling, due to the anharmonic terms in the potential energy surface, is pronounced at lower temperature and higher tetrahedrality. This finding was related to the electric field caused by the surrounding molecules and its positive correlation to the tetrahedral order parameter. Overall, the observed change in the mechanical anharmonic coupling could impact our understanding of the OH stretch relaxation mechanism. Although this relaxation has been largely assigned to the coupling with the intramolecular bending overtone, the intermolecular, hydrogen-bond modes are also believed to play an important role[34–36,88]. Strong mechanical anharmonic coupling between intramolecular stretching and intermolecular hydrogen-bond stretching modes could shorten the excited-state lifetime of the OH stretch. Such correlation would also agree with the experimentally observed trend of shorter OH stretch excitation lifetime at a lower temperature.

Finally, we observe that the $400\,\text{cm}^{-1} < \omega_1 < 1000\,\text{cm}^{-1}$ region of the IIR spectrum, corresponding to the librations of water molecules, carries rich information about liquid water and should be further explored experimentally. In particular, such studies could provide additional information on the combination bands appearing in IR absorption and Raman scattering spectra that have challenged physical chemists for decades.

## Methods

### Dipole moment and polarizability models

The induced dipole moments and polarizabilities were modeled as[44]

$$\boldsymbol{\mu}^{ind} = \sum_{\substack{i,j \\ i \neq j}} \boldsymbol{\Pi}_i \cdot \mathbf{E}_{ij}, \tag{8}$$

$$\boldsymbol{\Pi}^{ind} = -\sum_{\substack{i,j \\ i \neq j}} \boldsymbol{\Pi}_i \cdot \mathbf{T}_{ij} \cdot \boldsymbol{\Pi}_j, \tag{9}$$

where $\mathbf{E}_{ij} = \sum_{a\in\{M,H1,H2\}} Q_{ja}\mathbf{r}_{ja,iO}/r_{ja,iO}^3$ is the electric field produced by molecule $j$ on the oxygen atom of molecule $i$, $\mathbf{T}_{ij} = \mathbf{T}(\mathbf{r}_{jO,iO})$, $\mathbf{T}(\mathbf{r}) = (r^2 - 3\mathbf{r}\otimes\mathbf{r})/r^5$ is the $3\times3$ dipole-dipole interaction tensor, $\mathbf{r}_{ja,iO} = \mathbf{r}_{ja} - \mathbf{r}_{iO}$, $\mathbf{r}_{ja}$ denotes the three-dimensional position vector of atom $a \in \{M, O, H1, H2\}$ of molecule $j$, $Q_{ja}$ is its partial qTIP4P/F charge, $r = \|\mathbf{r}\|$ denotes the norm of vector $\mathbf{r}$, and $\otimes$ is the outer product of two vectors. M denotes the M site of the TIP4P model, whose position is related to the molecular geometry of molecule $i$ by $\mathbf{r}_{iM} = \gamma\mathbf{r}_{iO} + (1-\gamma)(\mathbf{r}_{iH1} + \mathbf{r}_{iH2})/2$, where $\gamma = 0.73612$[66]. Permanent dipole moment of molecule $i$ was $\boldsymbol{\mu}_i = \sum_{a\in\{M,H1,H2\}} Q_{ia}\mathbf{r}_{ia}/c$, where $c = 1.3$ is a scaling factor that reduces the dipole moment of water to its gas-phase value[44,63]. The permanent polarizability of each molecule $i$, $\boldsymbol{\Pi}_i \equiv \boldsymbol{\Pi}_i(\mathbf{r}_{iO}, \mathbf{r}_{iH1}, \mathbf{r}_{iH2})$, was computed according to ref. 67 in the molecular reference frame and rotated into the laboratory frame. Then, the total dipole moment and polarizability are given by

$$\boldsymbol{\mu} = \sum_i \boldsymbol{\mu}_i + \boldsymbol{\mu}^{ind}, \tag{10}$$

$$\boldsymbol{\Pi} = \sum_i \boldsymbol{\Pi}_i + \boldsymbol{\Pi}^{ind}. \tag{11}$$

Finally, we note that the sums over $i$ in all expressions above are taken over the molecules in a unit cell, while the sums over $j$ extend to infinity

beyond the central unit cell. Due to the slow convergence of Coulomb interactions in the direct space, we used Ewald summation for the electric field $\mathbf{E}_{ij}$ and tensor $\mathbf{T}_{ij}$ [22,63].

## Computational details

The classical thermal average of Eq. (2) in the main text was evaluated by sampling from 5 independent NVT trajectories, each equilibrated for 100 ps, of a water box with 64 molecules and the cell parameter adjusted to the experimental density at any given temperature. Initial structures were taken from ref. 24. A time step of $\Delta t = 0.25$ fs, second order symplectic integrator, and a Langevin thermostat with the time constant $\tau = 100$ fs were used throughout. From 2560 initial samples collected in this way, we launched 25 ps NVE equilibrium trajectories, along which we collected the dipole moments and polarizabilities every 4 steps (1 fs) and the derivatives of the dipole moment every 1000 steps (250 fs). Finally, starting from positions at which the dipole derivatives were evaluated, "nonequilibrium" trajectories, i.e., with modified initial momenta $\mathbf{p}_{\pm,0}$, were propagated for 1000 steps (250 fs). This generated $2.56 \times 10^5$ samples for Eq. (2) of the main text, which is comparable to the numbers used in Refs. 46,62,63, and the response function was computed for 250 fs in both $t_1$ and $t_2$. The statistical error was estimated using bootstrapping and is shown in Supplementary Fig. 1. We set $\varepsilon = 2$, which converts into the electric field magnitude of $E_0 = \varepsilon/\Delta t \approx 10$ V/Å. A weaker electric field was also tested ($\varepsilon = 0.2$a.u., see Supplementary Fig. 2) in a simulation with the doubled number of initial conditions.

RPMD simulations were performed in a similar way, using 4 independent thermostatted path-integral MD (PIMD) trajectories to produce 512 samples from which TRPMD trajectories were launched. Each TRPMD equilibrium trajectory was propagated for 50 ps (200000 steps) and nonequilibrium trajectories was launched every 1000 steps. In total, this resulted in $1.024 \times 10^5$ initial conditions. PIMD trajectories used a path-integral Langevin equation (PILE) thermostat for the normal modes with the centroid coupled to a global velocity rescaling thermostat with $\tau = 100$ fs[89], whereas the TRPMD trajectories used a generalized Langevin equation (GLE) thermostat coupled to the normal mode representation of the ring polymer[74].

Linear absorption and anisotropic Raman spectra were computed from the equilibrium trajectories according to:

$$I^{\text{IR}}(\omega) \propto -\omega \text{Im} \int_0^\infty \langle \boldsymbol{\mu}(t) \cdot \dot{\boldsymbol{\mu}}(0) \rangle e^{-t^2/2\sigma_t^2} e^{-i\omega t} dt, \qquad (12)$$

$$I^{\text{Raman}}(\omega) \propto -A(\omega) \text{Im} \int_0^\infty \langle \text{Tr}[\boldsymbol{\beta}(t) \cdot \dot{\boldsymbol{\beta}}(0)] \rangle e^{-t^2/2\sigma_t^2} e^{-i\omega t} dt, \qquad (13)$$

where $\boldsymbol{\mu}$ is the dipole moment vector of the cell, $\boldsymbol{\beta}(t) = \boldsymbol{\Pi}(t) - \mathbf{I}\text{Tr}[\boldsymbol{\Pi}(t)]/3$, $\boldsymbol{\Pi}(t)$ is the polarizability tensor, $\mathbf{I}$ is the $3 \times 3$ identity matrix, and $\sigma_t = 1000$ fs. $A(\omega)$ was different for the two Raman experiments shown in Fig. 1. For the broad-range Raman spectrum shown in Fig. 1b, $A(\omega) = [1 - \exp(-\beta\hbar\omega)]^{-1}$[27], while for the low-frequency part shown in the inset of that figure, which was experimentally measured by optical Kerr effect[16], $A(\omega) = 1$.

Two-dimensional IIR spectra were computed from the response function according to Eq. (3) of the main text, but with a damping $f(t_1,t_2) = \exp(-(t_1+t_2)^{12}/\tau_{12})$, $\tau_{12} = 5 \times 10^{28}$ fs$^{12}$, applied prior to taking the discrete sine transform. We used the $z$-component of the dipole moment and $zz$-component of the polarizability, resulting in the $zzzz$-component of the response function. The time-dependent response function is computed in the atomic units of [length]$^3 \times$ [dipole]$^2$/ ([energy] $\times$ [time])$^2$ and the corresponding spectra in the atomic units of [length]$^3 \times$ [dipole]$^2$/[energy]$^2 \equiv e^2 a_0^5/E_h^2$.

## Data availability

The data generated in this study, together with the input files and scripts for plotting the data, have been deposited in the Zenodo database under accession code DOI:10.5281/zenodo.7619094.

## Code availability

Modified i-PI code[90] used to run MD and TRPMD simulations is available at https://github.com/tbegusic/i-pi.git and https://doi.org/10.5281/zenodo.7682877; `encorr` code used for processing the outputs is available at https://github.com/tbegusic/encorr.git and https://doi.org/10.5281/zenodo.7682972.

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

## Acknowledgements

The authors thank Haw-Wei Lin, Roman Korol, and Vignesh C. Bhethanabotla for helpful discussions. T.B. acknowledges financial support from the Swiss National Science Foundation through the Early Postdoc Mobility Fellowship (grant number P2ELP2-199757). The authors gratefully acknowledge support from the National Science Foundation Chemical Structure, Dynamics and Mechanisms program (grant CHE-1665467). The computations presented here were conducted in the Resnick High Performance Computing Center, a facility supported by Resnick Sustainability Institute at the California Institute of Technology.

## Author contributions

T.B. conceived the study, implemented the theoretical models and methods, performed the simulations, and analyzed the data; T.B. and G.A.B. discussed the results and wrote the manuscript.

## Competing interests

The authors declare no competing interests.
