## [Peer Review File · Nature Communications]

Two-dimensional infrared-Raman spectroscopy as a probe of water's tetrahedralityREVIEWER COMMENTS

Reviewer #1 (Remarks to the Author):

This is a very solid paper that should eventually be published. It uses state-of-the-art MD simulations to simulate the 2D spectroscopic response of water, covering its inter- as well as intra-molecular modes. The paper focuses on the IIR spectrum, an experiment pioneered by the Blake group, but the simulations could easily be adapted to also simulate other hybrid multidimensional techniques performed by other groups. Simulations of that sort, as well as the corresponding experiments, are very difficult and thus still rare in the literature.

Having said that, there is not too much that is conceptually new in this paper. The work that is probably the closest is Ref. 44. The present paper goes beyond Ref. 44 by exploring nuclear quantum effects, and by relating the temperature dependence of the response to the tetrahedrality of water (I will comment on the latter below). Nuclear quantum effects are conceptually new, and to the best of my knowledge has not been explored in the context of this type of experiments. But unfortunately (for the authors), the conclusion is that nuclear quantum effects don't play a big role. But it is still important information that this is the case. Furthermore, we don't really learn anything new about water (yet). The concrete experiment that this paper simulates does not yet exist, and probably will be very hard. The experiment that comes the closest to the present discussion is that of Ref. 39. In that regard, I find it quite disappointing that the present paper does not really discuss the connection to that experiment (it is only briefly mentioned in Supp). So, overall, that is a very solid paper, that should be published after revision. However, at this point, it will probably be interesting to only a relatively small specialized audience.

Some more concrete comments:

- My biggest concern is Fig. 4a, and the conclusion drawn from it. The conclusion that a "isosbestic can be attributed to the existence of two components, whose populations change with temperature" is highly controversial, to say the least. See e.g. JPC B 2015 119, 8406 and JACS 2005, 7, 14930, both of which argue against that conclusion from very different perspectives.

- Furthermore, I don't really understand the assignment of the difference between electrical vs mechanical anharmonicity. It is done mostly by reference to Ref. 44, and a black-box-simulation in Supplementary Information. But what is the physical picture? Much of the discussion is on these two types of anharmonicities, so a bit more of an explanation/discussion would really be needed.

- Related to that, why would mechanical anharmonicity dominate the inter/intramolecular coupling of "tetrahedral water", and electrical anharmonicity that of "unstructured water"? Do the authors have any physical picture/explanation for that difference?

- Why is the combination transition at 2150 cm^{-1} missing in the 1D spectra, despite the fact that the simulation is anharmonic. I'm pretty sure that other MD can reproduce that band. What is the

difference here? It is strange that a corresponding cross peak is nevertheless observed. It seems that the authors struggle with that as well, as the discussion remains very open.

- In the conclusion it is said that Ref. 39 measured “only” the absolute value 2D spectrum, and consequently don’t really discuss it further. Do the authors want to say that absolute value basically kills all the spectroscopic features discussed here? Why? I’m pretty sure the Bonn group has recently also performed a heterodyned version of the same experiment, but I’m not whether that is already published.

Minor comments:

- On page 4, it is said that the way to describe the polarizability is “sufficiently accurate”. How would you ever know whether that is really the case, in particular as there is no experiment to compare with?

- In Eq.2, I didn’t understand what epsilon is. If it is the field, shouldn’t it be in the numerator? But shouldn’t it be a field³? Why do we need it at all?

- The second rectangle is missing in Fig. 2

Reviewer #2 (Remarks to the Author):

The authors have done a good contribution to the field of theoretical hybrid 2DIR. The modelling of such spectra is challenging, and in my view the authors show a quite good set of results, such as the contribution of quantum effects for the spectra.

These results have a significant contribution to either experimental and theoretical spectroscopy of condensed matter systems. Furthermore, this work paves the way to developed higher accuracy methods for spectroscopy.

The authors should mention more previous studies from 2DIR of the OH stretch not only for water but for other systems, for example from Jim Skinner, and Peter Hamm. Note that, these systems such as ICE or alcohols should have the same effects mentioned in this paper. Thus, a stronger link to previous work should be mentioned.

The conclusions should be supported by stronger data analysis, for example rdf of water with the FF used, with a comparison to other water models. Also a stronger link to structural features should be done.

There is enough detail to reproduce the results shown in the paper. Also the results presented in the paper meet the standards of the field.

Reviewer #1 (Remarks to the Author):

This is a very solid paper that should eventually be published. It uses state-of the art MD simulations to simulate the 2D spectroscopic response of water, covering its inter- as well as intra-molecular modes. The paper focuses on the IIR spectrum, an experiment pioneered by the Blake group, but the simulations could easily be adapted to also simulate other hybrid multidimensional techniques performed by other groups. Simulations of that sort, as well as the corresponding experiments, are very difficult and thus still rare in the literature.

Having said that, there is not too much that is conceptually new in this paper. The work that is probably the closest is Ref.44. The present paper goes beyond Ref. 44 by exploring nuclear quantum effects, and by relating the temperature dependence of the response to the tetrahedrality of water (I will comment on the latter below). Nuclear quantum effects are conceptually new, and to the best of my knowledge has not been explored in the context of this type of experiments. But unfortunately (for the authors), the conclusion is that nuclear quantum effects don't play a big role. But it is still important information that this is the case. Furthermore, we don't really learn anything new about water (yet). The concrete experiment that this paper simulates does not yet exist, and probably will be very hard. The experiment that comes the closest to the present discussion is that of Ref. 39. In that regard, I find it quite disappointing that the present paper does not really discuss the connection to that experiment (it is only briefly mentioned in Supp). So, overall, that is a very solid paper, that should be published after revision. However, at this point, it will probably be interesting to only a relatively small specialized audience.

Response: We thank the reviewer for the positive and stimulating comments, and hope that our point-by-point responses below and the changes made to the manuscript convince the reviewer that there are important new ideas brought forward by this work. In the revised manuscript, we further justify our assignment of spectral features to the mechanical/electrical anharmonic coupling between low- and high-frequency modes and provide a physical picture of the observed temperature- and tetrahedrality-dependent changes. These two points are discussed in more detail in the point-by-point responses below. Furthermore, we discuss our results in the context of the anomalous temperature dependence of the OH stretch relaxation dynamics in liquid water and ice. First, in the introduction (page 2, bottom, and page 3, top), where we discuss a more common 2DIR spectroscopy of high-frequency modes, we add:

"2DIR has been successfully applied to analyze coupling among high-frequency modes in ice³¹ and in proteins,³² but also to understand the vibrational relaxation mechanisms in liquids.³³ For example, several 2DIR studies on liquid water and ice speculated that mechanical anharmonic coupling to intermolecular modes contributes to the relaxation of the OH stretch.³⁴⁻³⁶ In addition, non-Condon effects, which are related to electrical anharmonic coupling between high- and low-frequency modes, were shown to affect the 2DIR spectra of OH stretching mode in liquid water.³⁷"

Second, in Discussion (page 16), we conclude:

"Overall, the observed change in the mechanical anharmonic coupling could impact our understanding of the OH stretch relaxation mechanism. Although this relaxation has been largely assigned to the coupling with the intramolecular bending overtone, the intermolecular, hydrogen-bond modes are also believed to play an important role.^{34-36,89} Strong mechanical anharmonic coupling between intramolecular stretching and intermolecular hydrogen-bond stretching modes could shorten the excited-state lifetime of the OH stretch. Such correlation would also agree with the experimentally observed trend of shorter OH stretch excitation lifetime at lower temperature."

In other words, our work is also an important step towards understanding the vibrational relaxation dynamics in liquid water, whose rate is, anomalously, increased at lower temperature. Importantly, the manuscript is not only a computational study of liquid water, but also a test of capabilities of an emerging type of hybrid THz/IR/Raman two-dimensional spectroscopies. Therefore, we not only present interesting and, in our opinion, conceptually new, ideas about water, but we also relate them directly to spectroscopic observables and try to answer the question: What can we learn from hybrid IR-Raman spectroscopy? To emphasize this, following the reviewer's suggestion, we now discuss the TIRV experiment in the main text (new Fig. 5 and paragraph on page 11). Further, we now cite recent work on recovering the complex-valued, frequency-dependent response function retrieved by TIRV spectroscopy, which is directly related to the results of our simulations. Although this latest work does not include results on water, it demonstrates that this missing experiment is within reach.

Some more concrete comments:

- My biggest concern is Fig. 4a, and the conclusion drawn from it. The conclusion that a "isosbestic can be attributed to the existence of two components, whose populations change with temperature" is highly controversial, to say the least. See e.g. JPC B 2015 119, 8406 and JACS 2005, 7, 14930, both of which argue against that conclusion from very different perspectives.

Response: Our results and interpretation of spectral features do not depend on the interpretation of water as a homogeneous or a two-state heterogeneous liquid. Fig 4a only shows that the probability distribution $P(Q)$ of the tetrahedral order parameter Q changes with temperature and that it exhibits an isosbestic point, which has been seen in other papers, including those proposed by the reviewer. Fig 4b shows that the spectra of high (low) Q water agree with the features seen at low (high) temperatures, i.e., we establish a correlation between Q and spectra and argue that this correlation is responsible for the temperature dependence of the spectra. The origins of the changes in $P(Q)$ with temperature are not discussed in our work, except for the sentence mentioned by the reviewer. We rewrite that part to (pages 11-12):

"The distribution of Q at thermal equilibrium exhibits a strong temperature dependence and a clear isosbestic point at $Q \approx 0.67$ (see Fig. 6a). In the studied temperature range, the bimodal distribution can be decomposed into two temperature-independent components whose populations change with temperature.²⁶ However, as shown by Geissler,⁷⁸ the appearance of an isosbestic point does not necessarily imply that water is a heterogeneous mixture. More recently, the increased population of low- Q water molecules at higher temperature has been assigned to the appearance of neighboring molecules in the interstitial position between the first and second solvation shells.⁷⁷"

- Furthermore, I don't really understand the assignment of the difference between electrical vs mechanical anharmonicity. It is done mostly by reference to Ref. 44, and a black-box-simulation in Supplementary Information. But what is the physical picture? Much of the discussion is on these two types on anharmonicities, so a bit more of an explanation/discussion would really be needed.

Response: It is difficult to separate the mechanical and electrical anharmonicity. The former is always present, unless substantial changes to the force field are made, which would then poorly perform in modeling water. The latter is also difficult to turn off because the polarizability is generally a nonlinear function of coordinates (dipoles can be easily reduced to the permanent contribution, which can be modeled as a linear function of coordinates). As discussed in the new paragraph on pages 7-8, we explain how a signal containing predominantly mechanical anharmonic pathways can be generated from a Raman-IR-IR (RII) spectrum simulation. Additional justification is provided in the new Supplementary section IV, where we also derive an expression for relating the computed signal to the

mechanical anharmonic contribution to the IIR spectrum, which is the main target of our work. Overall, the additional simulations serve as a further verification of the assignment of spectral features to mechanical versus electrical anharmonicity contributions .

- Related to that, why would mechanical anharmonicity dominate the inter/intramolecular coupling of “tetrahedral water”, and electrical anharmonicity that of “unstructured water”? Do the authors have any physical picture/explanation for that difference?

Response: This result is explained in the new paragraph on pages 13-14. We use results of Auer and Skinner, J. Chem. Phys. 128, 224511 (2008), who fitted the OH stretching frequency and dipole moment derivative as functions of the external electric field caused by the surrounding water molecules. They showed that the frequency follows a strong quadratic trend, while the dipole derivative is roughly linear in the electric field. We then demonstrate that these trends can explain why the mechanical anharmonicity increases with the electric field strength, which is also positively correlated to the tetrahedral order parameter and agrees with our observations from the simulated 2D spectra.

- Why is the combination transition at 2150cm⁻¹ missing in the 1D spectra, despite the fact that the simulation is anharmonic. I’m pretty sure that other MD can reproduce that band. What is the difference here? It is strange that a corresponding cross peak is nevertheless observed. It seems that the authors struggle with that as well, as the discussion remains very open.

Response: We added the following explanation in a new paragraph on page 8 (bottom) and 9 (top):

“This discrepancy can be explained by the fact that the fundamentals of the one-dimensional spectra follow harmonic selection rules, while the peaks in the two-dimensional spectra appear solely due to the anharmonic excitation pathways. Therefore, the combination bands in the one-dimensional spectra can be orders of magnitude weaker than the fundamentals and still exhibit strong off-diagonal peaks in the two-dimensional spectra. Indeed, we see that the diagonal peak at around (1650 cm⁻¹, 1650 cm⁻¹) is much lower in intensity than the off-diagonal libration-bending peak, implying that the anharmonicity within the bending mode is weaker than its anharmonic coupling to the intermolecular librations.”

We do not agree, however, with the statement that “other MD can reproduce that band”. In fact, we found that most models struggle as well to describe the band accurately and that the best results were attained with ab initio approaches, which are still beyond reach for our 2D spectra simulations. We therefore write this in the same paragraph as above and provide appropriate references:

“Let us also note that other force fields and dipole/polarizability models also heavily underestimate or fail to reproduce the combination band at 2150 cm⁻¹,^{19,21,48,70} unlike the ab initio approaches, which appear to systematically reproduce this band.^{26,71,72”}

- In the conclusion it is said that Ref. 39 measured “only” the absolute value 2D spectrum, and consequently don’t really discuss it further. Do the authors want to say that absolute value basically kills all the spectroscopic featured discussed here? Why? I’m pretty sure the Bonn group has recently also performed a heterodyned version of the same experiment, but I’m not whether that is already published.

Response: We now introduce in the main text (Fig. 5 and paragraph on page 11) the spectra that correspond to the experiment by the Bonn group. In contrast to the original submission, we now simulate the more recent spectrum of J. Chem. Phys. 154, 174201 (2021), where they could separate positive and negative ω_1 frequency. We show that the absolute value spectra indeed lose the

interesting temperature-dependent features because these features rely on a sign difference (the positive and negative lobes in the sine-sine spectra). Nevertheless, at the end of the new paragraph on page 11, we also note: “Therefore, an accurate determination of the full response function will be needed to experimentally measure the spectral features appearing in the sine-transform TIRV spectra. Fortunately, a scheme that could achieve this goal has been recently applied to the TIRV spectrum of liquid dimethyl sulfoxide,⁷⁴ demonstrating that similar experiments on water are within reach.” New Ref. 74 (Seliya, Bonn, Grechko, [arXiv:2212.05593](https://arxiv.org/abs/2212.05593), 2022) appeared online after our initial submission and adds to the relevance of our computational work.

Minor comments:

- On page 4, it is said that the way to describe the polarizability is “sufficiently accurate”. How would you ever know whether that is really the case, in particular as there is no experiment to compare with?

Response: We can only test models by comparing to available experiments or other simulations. In this work, we do both: (i) We compare the simulated 1D spectra to the experimental ones in Fig. 1 and (ii) we find that our 2D spectra, and in particular the interesting spectral features in the TIRV frequency region, agree with the simulations of Ito and Tanimura, J. Chem. Phys. 144, 074201 (2016), who used different force-field and dipole/polarizability models. In the revised manuscript, we delete the sentence (see page 5, bottom) “Nevertheless, our approach is sufficiently accurate for the analysis that is presented in this work.” and directly refer the reader to Fig. 1 for the comparison of 1D spectra. Apart from a blue-shifted and narrow OH stretching band (which we refer to in the new paragraph on page 11 where we compare simulated and experimental 2D TIRV spectrum), the main qualitative flaw of our simulated 1D spectra is the combination band at 2150 cm⁻¹, which we explained is not that uncommon for comparable force field/dipole/polarizability models (page 9, top).

- In Eq.2, I didn't understand what epsilon is. If it is the field, shouldn't it be in the numerator? But shouldn't it be a field^3? Why do we need it at all?

Response: The response function has no explicit dependence on the external fields. Here, the epsilon parameter appears in two places – in the prefactor and in the modified initial momenta of the nonequilibrium trajectories (p+ and p-). For sufficiently small epsilon, the final result is independent of the explicit value of epsilon, which is what we expect to have for the response function. However, finite epsilon is often used because it improves the statistical convergence with respect to the number of trajectories. The method has been discussed in several papers which are cited above equation (2). In the revised manuscript, we write:

“ ϵ is a free parameter in the calculations and corresponds to the magnitude of the external electric field integrated over the short interaction time. For sufficiently small ϵ , the thermal average of equation (2) is linear in ϵ , meaning that the time-dependent response function $R^{MD}(t_1, t_2)$ is, as expected, independent of its exact value.”

To verify that our ϵ is sufficiently small, in the original submission we already included a comparison between spectra simulated with two different values of ϵ (Supplementary Fig. 2).

- The second rectangle is missing in Fig. 2

Response: We made the square borders thicker to make sure they are visible in different PDF viewers or in print.

Reviewer #2 (Remarks to the Author):

The authors have done a good contribution to the field of theoretical hybrid 2DIR. The modelling of such spectra is challenging, and in my view the authors show a quite good set of results, such as the contribution of quantum effects for the spectra.

These results have a significant contribution to either experimental and theoretical spectroscopy of condensed matter systems. Furthermore, this work paves the way to developed higher accuracy methods for spectroscopy.

The authors should mention more previous studies from 2DIR of the OH stretch not only for water but for other systems, for example from Jim Skinner, and Peter Hamm. Note that, these systems such as ICE or alcohols should have the same effects mentioned in this paper. Thus, a stronger link to previous work should be mentioned.

The conclusions should be supported by stronger data analysis, for example rdf of water with the FF used, with a comparison to other water models. Also a stronger link to structural features should be done.

There is enough detail to reproduce the results shown in the paper. Also the results presented in the paper meet the standards of the field.

Response: We appreciate the reviewer's positive comments and suggestions.

In the revised manuscript, page 3 (top), we include additional references to the work on 2DIR spectroscopy. We relate our work to previous attempts in the 2DIR community, including Hamm and Skinner, to address the vibrational relaxation mechanism of the intramolecular OH stretch mode. In the introduction (page 3, top), we write: "*2DIR has been successfully applied to analyze coupling among high-frequency modes in ice and in proteins, but also to understand the vibrational relaxation mechanisms in liquids. For example, several 2DIR studies on liquid water and ice speculated that mechanical anharmonic coupling to intermolecular modes contributes to the relaxation of the OH stretch.*"

In our work, we demonstrated that the mechanical anharmonic coupling between high- and low-frequency modes increases as the temperature decreases. In the revised manuscript (page 15, bottom, and 16, top), we therefore conclude:

"Overall, the observed change in the mechanical anharmonic coupling could impact our understanding of the OH stretch relaxation mechanism. Although this relaxation has been largely assigned to the coupling with the intramolecular bending overtone, the intermolecular, hydrogen-bond modes are also believed to play an important role.^{34-36,89} Strong mechanical anharmonic coupling between intramolecular stretching and intermolecular hydrogen-bond stretching modes could shorten the excited-state lifetime of the OH stretch. Such correlation would also agree with the experimentally observed trend of shorter OH stretch excitation lifetime at lower temperature."

Furthermore, as discussed in a response to Reviewer 1, to explain the relation between mechanical anharmonic coupling and tetrahedrality, we use the work of Skinner and explain our results in terms of the dependence of OH stretch frequency and dipole moment derivative on the electric field exerted by the surrounding water molecules (pages 13-14).

In our work, we used a well-established force field, which was benchmarked against experimental properties of liquid water, including radial distribution functions. To emphasize this, in the revised manuscript we write:

"To model water, we used a flexible, point-charge qTIP4P/F force field,⁶⁶ which has been well studied for spectroscopic simulations²⁵ and benchmarked against a number of experimental thermodynamic properties of water, including the radial distribution functions, dielectric constant, density, and melting point. As a point-charge force field, the model is computationally efficient, which is needed for

statistically averaging two-time response functions. In addition, because it is flexible, we could study all vibrational degrees of freedom, including high-frequency intramolecular modes."

Further, to clarify that our choice of dipole/polarizability functions is not related to the force field, we write later:

"The introduced induced dipole and polarizability effects were not used in the evaluation of the forces, which were computed according to the original, non-polarizable qTIP4P/F model."

Below we provide point-by-point replies to specific comments shared in the annotated PDF:

1. Page 3, top: *"The THz-THz-Raman and TIRV methods are related to other two-dimensional IR-Raman techniques, namely the two-dimensional IR doubly vibrationally enhanced⁴² and IR-IR-visible sum-frequency generation spectroscopies.⁴³"*

There is a lot of work done in this field. Please add more citations.

Response: We added more citations.

2. Page 4 (under Eq. 3): *"To model water, we used a fast and accurate qTIP4P/F force field..."*

- More information about this model should be given. Such as the rdfs of this water model, and how they compare to experiment.

Response: This paragraph was rewritten to clarify that we use an established force field that has been documented in the original reference Habershon, Markland, Manolopoulos, J. Chem. Phys. 131, 024501 (2009). There, the radial distribution functions were already benchmarked against the experimental results.

- This usually does not go hand in hand. What does accurate means in this context? Why is this FF more accurate than others? It has been shown in previous studies that other FF's reproduce quite well the spectra of water. Why is this one more accurate?

Response: We do not claim that the force field or dipole/polarizability models we used are more accurate than others. Our simulations impose a set of limitations on these choices. For example, to converge two-time response function calculations, we must use a computationally efficient force field, which led us to computationally affordable point-charge models. We wanted to study intramolecular dynamics as well, which meant that the force field had to be flexible (unlike standard TIP and SPC models, that were constructed for rigid water simulations). Finally, to study nuclear quantum effects, the force field must not be fitted against experiment using MD simulations, otherwise the quantum effects can be overestimated. qTIP4P/F satisfies these conditions. We now discuss each of these points on pages 3 (bottom), 4, and 5 (top). On page 5 (bottom), we acknowledge that other models for simulating spectra exist but also write:

"However, while some of them were not readily available, others considered only rigid water molecules or were fitted to experiments using classical MD simulations, which would prevent us from accurately exploring nuclear quantum effects."

3. Page 5, Fig. 1

The authors should also refer to the work of Jim Skinner, which has been a pioneer in calculating the IR spectra for the OH stretch.

Response: In the revised manuscript, we refer to the work of Skinner in several places. Most importantly, we use some of his results to explain the relation between the mechanical anharmonic coupling and tetrahedral order parameter, which is one of the central results of our manuscript (see new paragraph on pages 13-14).

4. Page 5, caption of Fig. 1 *“IR absorption (top) and anisotropic Raman (bottom) spectra of liquid water at 300 K simulated with the classical MD and TRPMD, compared with the experiments of Refs. 16,27,54. The inset in the bottom panel compares the low-frequency part of the simulated and experimental anisotropic Raman spectra.”*

How do these results compare to simulations done with other water models, such as SPC/E and the TIP's from Carlos Vega.

Response: Most of the SPC and TIP force fields are rigid, which is why they cannot be used to produce full IR or Raman spectra through MD or RPMD simulations. In the revised manuscript, we explain that we need a flexible force field on page 5 (top).

5. Comment on *“...we have developed a new RPMD approach,⁶¹ which reproduces correctly the quantum-mechanical thermal distribution and preserves it during approximate classical dynamics.”* The correct quantum mechanical thermal distribution at which level of theory? That is not clear to me from the text.

Comment on *“... reproduces exact quantum-mechanical thermal distribution...”*

Which level of theory? DFT, CC, CI?

Response: “Exact quantum-mechanical thermal distribution” refers to the nuclear (vibrational) degrees of freedom for a given potential energy surface. To clarify this, in the revised manuscript, we shorten the first sentence (page 6, starting with “Recently, we have developed...”):

“Recently, we have developed a new RPMD approach,⁷² which can simulate, at least approximately, the nuclear quantum effects on two-dimensional IIR spectra.”

And then later we write:

“For a given potential energy surface, the extended classical system in the limit of $N \rightarrow \infty$ reproduces exact quantum-mechanical thermal distribution of nuclear degrees of freedom, while if $N = 1$, RPMD reduces to classical MD.”

We hope that this clarifies that RPMD only attempts to include quantum effects of nuclear degrees of freedom and has no direct connection to the accuracy or exactness of the potential energy surface (which is related to the electronic structure problem and the quantum chemistry methods implied by the Reviewer).

7. Comment on *“In the low frequency region of interest, we observe a strong peak at about (600, 1650) to anharmonic coupling between librations (400-1000) and intramolecular bending mode.⁶⁵ The same coupling mechanism is responsible for the combination transition at 2150 cm^{-1} , around also known as the “association” band in the one-dimensional spectra. Interestingly, this combination band is not captured in our model (see Fig. 1), even though the corresponding two-dimensional spectral feature clearly appears in the IIR spectrum.”*

How does this relate to the structure of water? The authors used a force field, and did MD simulations at different temperatures. However, I don't see any analysis done on those MD simulations. I don't know how the FF compares to others, nor to the experimental results of water.

Response: The structural properties of water, such as the radial distribution functions, were benchmarked against the experimental results in the original work that introduced the qTIP4P/F force field (Habershon, Markland, Manolopoulos, J. Chem. Phys. 131, 024501 (2009)). As seen from Figs. 1-3 of the said reference, the results clearly agree well with the experimental radial distribution functions. For this reason, we believe that the lack of the combination band in our 1D spectra is due to the dipole and polarizability models, which could be further improved to include more subtle induced electrostatic or few-body terms. At the end of the paragraph on page 9 (top), we write:

“Since the force field we used reproduces the structural and dynamical properties of liquid water rather accurately,⁶⁶ we tentatively assign the absence of this combination band in the simulated spectra (Fig. 1) to the limitations of our dipole and polarizability models.”

8. Comment on: *“To directly correlate the local tetrahedral order parameter with the spectral features observed in the two-dimensional IIR spectra, we decomposed the spectrum into contributions of individual molecules (see Supplementary Section VII). Then, we simulated the spectra (see Fig. 4b) originating predominantly from the molecules with either high ($Q > 0.72$) or low ($Q < 0.62$) local tetrahedral order parameter.”*

It would be nice to include a direct relation between the structure of water and the spectra.

Response: In the results section on “Spectral signatures of water’s tetrahedrality in the IIR spectrum”, we have directly related the microscopic structure of water (measured by the local tetrahedral order parameter Q) and the spectral features in the TIRV spectrum. From these results, shown in Fig. 6, we conclude that the two are correlated, namely, that the one type of spectral features (originating from mechanical anharmonic coupling) appears predominantly for high- Q (tetrahedral) water molecules, whereas the electrical anharmonic coupling dominates for low- Q (non-tetrahedral) molecules. Therefore, we can establish a direct relation between spectral features and microscopic structure of water, and also explain it based on the origins of anharmonic coupling between hydrogen-bond intermolecular and OH intramolecular modes. This was available already in the original submission, and was further substantiated by analysis related to Fig. 3 (page 7, bottom, and page 8) and by explanation of this relationship on pages 13 (bottom) and 14 (top).

9. Comment on: *“We envision that this structure-spectrum relationship could also be used to probe the structure of complex aqueous systems, such as confined water and aqueous solutions.”*

Not clearly shown by the results...

Add more data that corroborates this statement.

Response: We combine this sentence with the one after to write (page 15):

“Further computational work is needed to confirm whether IIR spectra of related systems, such as aqueous solutions, alcohols, or ice, can be broadly mapped to the local tetrahedral order parameter.”

10. Comment on: *“Classical thermal average of equation (2) of the main text was evaluated by sampling from 5 independent NVT trajectories, each equilibrated for 100 ps, of a water box with 64 molecules and the cell parameter adjusted to the experimental density at any given temperature. Initial structure was taken from Ref. 24. A time step of $t = 0.25$ fs, second order symplectic integrator, and a Langevin thermostat with the time constant $t = 100$ fs were used throughout.”*

More analysis of these MD simulations are required.

Response: The parameters used in our simulations (such as the timestep, box size, and thermostat) are fairly common and have been validated in many different calculations. The force field, as discussed above, has also been validated. The choice of the dipole moment and polarizability functions, defined in the Methods section, has been validated by comparison with the experimental IR and Raman spectra. The most important remaining analysis was the convergence of 2D spectra with respect to the number of trajectories (shown in Supplementary Fig. 1) and with respect to the choice of the ϵ parameter of the equilibrium-nonequilibrium MD method for 2D spectroscopy (see equation (2) and comparison for two values of ϵ shown in Supplementary Fig. 2). For these reasons, we believe that sufficient analysis is provided regarding the MD and TRPMD simulations.

REVIEWERS' COMMENTS

Reviewer #1 (Remarks to the Author):

The authors have adequately addressed my and the other referee's comments, and I now recommend publication of the paper

Reviewer #2 (Remarks to the Author):

The authors have addressed all the points of the referee appropriately.